# The Brittleness of Priors: An Empirical Case for Adaptive Neural Architectures

Dawud Hage[1]

[1]Independent Researcher

## Abstract

Deep and compositional architectures enjoy strong theoretical advantages, yet practical models still hinge on *fixed architectural priors* whose alignment with data is rarely scrutinized. We report a rigorous isoparametric study comparing fully-connected ReLU MLPs, residual MLPs, and periodic-activation networks (SIREN) on canonical 1D function approximation. To isolate architecture from capacity, we employ a strict isoparametric control, matching total parameter counts across all models; to avoid memorization, we inject Gaussian noise, hold out validation data, apply early stopping, and evaluate both interpolation and extrapolation error. Our results reveal a fundamental *brittleness of priors*: no single static architecture dominates. Shallow-and-wide MLPs excel on smooth, non-compositional targets; deeper MLPs win on compositional signals; periodic activations are decisively superior on oscillatory targets; and residual connections, despite optimization benefits, underperform when their identity-preserving bias misaligns with the target family. These empirical trends cohere with theory on depth-enabled expressivity and spectral bias. We conclude that maximal generalization arises only when architectural bias matches data structure, motivating a shift from static design towards heterogeneous modular systems and, in the longer term, *Architecturally Plastic Networks* (APNs).

## 1 Introduction

The expressive power of depth and composition is well established: deep networks can approximate certain function classes exponentially more efficiently than shallow ones [1, 2]. At the same time, empirical studies reveal a *spectral bias*, showing that standard networks fit low-frequency structure first and struggle with high-frequency content [3]. Specialized priors—for example sinusoidal activations (SIREN) and Fourier features—can invert this behavior by giving the network a bias toward high-frequency structure [4, 5]. We term the strong dependence of generalization on this match between architectural bias and data structure the *brittleness of priors*. This work contributes a controlled, foundational study demonstrating that when model capacity is held fixed, a misalignment between an architecture's prior and the data's underlying structure severely degrades generalization.

## 2 Isoparametric Experimental Design

To fairly evaluate architectural priors, we designed a rigorous experimental protocol. We compare three architectural families: (i) plain ReLU MLPs of varying depth, (ii) residual MLPs with matching depth, and (iii) periodic-activation networks (SIREN). These models are trained to approximate three canonical 1D function classes: (a) smooth non-compositional polynomials (e.g., $x^2$), (b) purely oscillatory signals (e.g., $\sin x$), and (c) mixed-compositional functions (e.g., $\sin x + x^2$).

A core principle of our study is a strict *isoparametric control*: matching the total number of trainable parameters across all models—to ensure that performance differences are due to architectural priors, not model capacity. To ensure models learn the underlying function rather than memorizing noisy data, we employ a robust training procedure. We sample points on a fixed interval, add Gaussian noise to the training set, hold out a clean validation split, and train with Adam, using early stopping on the validation MSE to select the best model. We report mean±std error across multiple random seeds, evaluating both in-domain (interpolation) and out-of-domain (extrapolation) performance against the true, noise-free function.

## 3 Results and Key Findings

Our experiments yield three consistent findings. First, *no single architecture is universally optimal.* Different architectural motifs excel on different targets: shallow-and-wide MLPs dominate on smooth polynomials; deeper MLPs are superior for compositional or high-variation signals; and SIREN is decisively best on oscillatory targets. This directly confirms the brittleness of static priors.

Second, *residual bias can misalign.* Despite their well-known optimization advantages, residual MLPs consistently underperform their plain MLP counterparts when learning these functions from scratch.

This suggests that ease of optimization does not compensate for a fundamentally mismatched inductive bias when the target function is not an identity-like mapping.

Third, *extrapolation rankings are consistent.* The performance advantages observed during interpolation largely carry over to out-of-domain evaluation. For instance, periodic priors maintain their superiority on oscillatory targets beyond the training interval. These empirical patterns align with established theory on depth and composition [1, 2], spectral-bias analyses [3], and network expressivity characterizations [6, 7].

## 4 Discussion

Our evidence that generalization hinges on prior-data alignment has direct implications for current SOTA paradigms. Mixture-of-Experts (MoE) systems, for instance, typically deploy dozens of architecturally homogeneous experts [8]; our results argue compellingly for using a *heterogeneous* toolkit of experts that can cover a more diverse set of inductive biases.

This finding also informs the trajectory of Neural Architecture Search (NAS). The field has moved beyond simple design-time searches, with innovations like one-shot supernets [9] and hypernetwork-based weight sharing [10] making the search process more efficient. The frontier has pushed further into dynamic, training-time paradigms, with sophisticated methods like Bayesian Population-Based Training that co-evolve an entire population of models, jointly optimizing their weights and hyperparameters throughout a single training run [11]. Our work provides the empirical justification to take the next logical step: moving from these advanced training-time methods to a fully dynamic, *inference-time architectural adaptivity.* The pronounced performance gaps we observed suggest that a model capable of selecting the correct architectural motif for a given input could achieve superior performance and efficiency compared to any single, static model, however well-optimized.

This motivates a clear path forward, as laid out in established roadmaps for the field [12]. We propose a "Chimera" benchmark—a synthetic regime-switching time series—as a concrete testbed to evaluate and drive progress on this front.

## 5 Conclusion

A controlled, isoparametric comparison reveals that architectural priors are powerful but brittle. Maximal generalization arises only when these priors align with data structure. This foundational finding serves as a robust proof-of-concept, justifying a shift in focus from designing single, static architectures towards creating adaptive systems that can leverage architectural diversity. The long-term goal this work motivates is the development of *Architecturally Plastic Networks* (APNs): models that learn to adapt not only their weights but also their own structure in response to data.

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
