# OpenReview forum: "The Brittleness of Priors: An Empirical Case for Adaptive Neural Architectures"
_NLDL.org/2026/Abstracts_Track — NLDL 2026 Abstracts_

### Official Review · Reviewer_TYLu · 2025-10-30

**Soundness:** 3
**Correctness:** 2
**Rating:** 4
**Confidence:** 3

**Summary:**

The author's abstract effectively highlights the importance of understanding how deep networks approximate underlying patterns and how different priors can influence this process. I find it a really interesting read as it presents an insightful discussion showing that, even under isoparametric conditions (i.e., models that use the same parameters), varying priors can lead to brittle (inconsistent behaviour). This emphasises the need to better align priors with the data being modeled. Overall, the work is a good way to start conversations and future work by encouraging the development of more adaptable and robust learning methods.

**Strengths:**

I think it brings to light problems that are underexplored in the field, and their experiments and findings are well explained to me in the limited page allocation they had.

**Weaknesses:**

I think it does need to be tested on more data and a variety of other applications, and perhaps additional metrics on performance can be implemented.  However, it is an abstract so it's understandable that it has not been done.

---

### Official Review · Reviewer_UaQ2 · 2025-11-01

**Soundness:** 2
**Correctness:** 2
**Rating:** 2
**Confidence:** 4

**Summary:**

The paper presents an isoparametric study comparing plain MLPs, residual MLPs, and periodic-activation networks (SIRENs) on 1D function approximation tasks. The authors demonstrate that different architectures excel on different target functions (smooth polynomials, oscillatory signals, and compositional functions), arguing that no single static architecture is universally optimal. They conclude that architectural priors must align with data structure for optimal generalization, motivating a shift toward adaptive systems and Architecturally Plastic Networks (APNs).

**Strengths:**

**TL;DR**: The paper employs rigorous isoparametric controls to isolate architectural effects from capacity, and the experimental design with noise injection, validation splits, and early stopping demonstrates methodological care appropriate for an extended abstract.

**Long Form**:
- For an extended abstract format, the scope of architectural families and experimental conditions considered is reasonably comprehensive.
- The motivation for heterogeneous MoE systems and the proposed "Chimera" benchmark provide concrete next steps
- The isoparametric comparison framework is well-designed and appropriate for isolating the effect of architectural priors from model capacity
- The experimental protocol (Gaussian noise injection, validation splits, early stopping) shows attention to avoiding memorization and ensuring fair comparisons
- The connection to existing theory (depth expressivity, spectral bias) provides useful context

**Weaknesses:**

**TL;DR**: The paper presents no experimental results whatsoever, making it impossible to evaluate the claimed findings, and the toy-level architectures and trivial function classes severely limit the generalizability and novelty of conclusions that largely restate known principles.

**Long Form**:
- (major) No experimental results are shown—only claims about findings with references to a non-existent full paper. This is the most critical flaw. Even in an extended abstract format, sufficient space exists to include key plots or tables demonstrating the claimed brittleness of priors
- (major) The considered functions (x², sin x, sin x + x²) and shallow architectures (simple MLPs, residual MLPs, SIRENs) are far too basic. Without theoretical guarantees, empirical claims require demonstration on non-trivial problems
- (major) The finding that "no single architecture is universally optimal" is essentially a restatement of the no-free-lunch theorem. The observations about shallow-wide vs. deep architectures on smooth vs. compositional functions are well-established in the literature
- (major) The leap from 1D toy functions to claims about MoE systems and real-world applications is unjustified. Modern MoEs operate on high-dimensional latent representations from complex encoders—the dynamics in such settings may bear no resemblance to 1D function fitting
- (major) The claim that residual connections "underperform when their identity-preserving bias misaligns with the target family" ignores that ResNets were designed for very deep networks and optimization at scale, not shallow networks on trivial regression tasks
- (minor) References are formatted inconsistently (varying inclusion of DOIs, URLs, page numbers)
- (minor) The term "priors" is used loosely throughout when "inductive biases" would be more accurate
- (minor) The connection between toy 1D experiments and proposed applications (heterogeneous MoEs, APNs) requires substantially more justification

---

### Official Review · Reviewer_WAAm · 2025-11-03

**Soundness:** 3
**Correctness:** 3
**Rating:** 4
**Confidence:** 3

**Summary:**

This study presents an empirical approach for showing that no single static architecture dominates others on some specific problems. For the experiments, three different ML models are benchmarked against each other to approximate three different 1D functions. I believe that the future plans in connection with the study are worth discussing in the NLDL conference.

**Strengths:**

- The robust training protocol to reduce the risk of overfitting
- The study fucses on forming the ground of a promising approach, namely Architecturally Plastic Networks (APNs), that will be proposed in the future.
- Since this an abstract study, it is good to highlight what the future research directions may be. And this is addresses clearly in the submission.

**Weaknesses:**

- The problem definition in the study may not sound interesting to the community as it reads like what is already stated by the famous No Free Lunch theorem. In other words, the first finding, "no single architecture is universally optimal", is something already known.

---

### Decision · Program_Chairs · 2025-11-05

**Decision:**

Accept

**Comment:**

The reviewers found the abstract borderline, yet the PCs believe it will be of interest to the community and should have the opportunity be presented.